# TFEB regulates murine liver cell fate during development and regeneration

Nunzia Pastore [1,2✉], Tuong Huynh[1,2], Niculin J. Herz[1,2], Alessia Calcagni'[1,2], Tiemo J. Klisch[1,2], Lorenzo Brunetti [3,4], Kangho Ho Kim[5], Marco De Giorgi[6], Ayrea Hurley[6], Annamaria Carissimo[7], Margherita Mutarelli[7], Niya Aleksieva [8], Luca D'Orsi[7], William R. Lagor[6], David D. Moore [5], Carmine Settembre[7,9], Milton J. Finegold[10], Stuart J. Forbes[8] & Andrea Ballabio [1,2,7,9✉]

It is well established that pluripotent stem cells in fetal and postnatal liver (LPCs) can differentiate into both hepatocytes and cholangiocytes. However, the signaling pathways implicated in the differentiation of LPCs are still incompletely understood. Transcription Factor EB (TFEB), a master regulator of lysosomal biogenesis and autophagy, is known to be involved in osteoblast and myeloid differentiation, but its role in lineage commitment in the liver has not been investigated. Here we show that during development and upon regeneration TFEB drives the differentiation status of murine LPCs into the progenitor/cholangiocyte lineage while inhibiting hepatocyte differentiation. Genetic interaction studies show that *Sox9*, a marker of precursor and biliary cells, is a direct transcriptional target of TFEB and a primary mediator of its effects on liver cell fate. In summary, our findings identify an unexplored pathway that controls liver cell lineage commitment and whose dysregulation may play a role in biliary cancer.

[1] Jan and Dan Duncan Neurological Research Institute, Texas Children Hospital, Houston, TX 77030, USA. [2] Department of Molecular and Human Genetics, Baylor College of Medicine, Houston, TX 77030, USA. [3] Stem Cells and Regenerative Medicine Center, Baylor College of Medicine, Houston, TX 77030, USA. [4] Center for Cell and Gene Therapy, Baylor College of Medicine, Houston, TX 77030, USA. [5] Department of Molecular and Cell Biology, Baylor College of Medicine, Houston, TX 77030, USA. [6] Department of Molecular Physiology and Biophysics, Baylor College of Medicine, Houston, TX 77030, USA. [7] Telethon Institute of Genetics and Medicine (TIGEM), Pozzuoli, NA 80078, Italy. [8] MRC Centre for Regenerative Medicine, University of Edinburgh, Edinburgh EH16 4UU, UK. [9] Department of Translational Medicine, Medical Genetics, Federico II University, Naples 80131, Italy. [10] Department of Pathology, Baylor College of Medicine, Houston, TX 77030, USA. ✉email: pastore@tigem.it; ballabio@tigem.it

The adult liver is the largest internal organ and provides many essential metabolic, exocrine, and endocrine functions[1]. Being a complex organ with several cell types, liver development involves multiple cell fate decisions. For instance, during development hepatic endoderm cells, known as hepatoblasts (HBs), differentiate into hepatocytes or cholangiocytes depending on their localization with respect to the portal vein: HBs exposed to ligands from portal venous endothelial cells differentiate into primitive ductal plate cells and then form bile ducts, whereas HBs located further away from the portal vein differentiate into hepatocytes[2].

Proliferation and cell fate choice are not only restricted to the embryonic stage. Indeed, despite hepatocytes in the adult liver rarely divide, under conditions in which alterations of liver mass occurs, such as surgical removal or cell loss caused by drugs or viruses, quiescent hepatocytes become proliferative and replicate to restore full liver functional capacity[3]. In certain injury models, hepatocytes can transdifferentiate into ductal biliary epithelial cells (BECs) to ensure tissue regeneration[4–8]. However, when hepatocyte proliferation is compromised, BECs become active and subsequently differentiate into hepatocytes[9,10]. Liver stem/progenitor cells (LPCs) may appear in chronic liver damage when hepatocyte proliferation is compromised and differentiate in both hepatocytes and bile ducts[11]. In humans, LPCs are evident in pathological ductular reactions often observed in a variety of liver diseases, including fatty liver diseases[12], chronic viral hepatitis[13], cirrhosis[14], and acute hepatic injury[15]. Thus, LPCs are considered potential targets for liver cell transplantation and repopulation[16]. However, although LPCs have capacity to differentiate into hepatocytes and biliary cells in vitro and to form hepatocyte buds repopulating the liver in vivo, their ability to participate to liver regeneration in human clinical setting is still unclear[17]. A specialized type of cells with both LPCs and mature hepatocytes features, called hybrid hepatocytes (HybHPs), express both SRY (sex determining region Y)-box (SOX9), a marker of LPCs, and the hepatocyte marker hepatocyte nuclear factor 4α (HNF4α)[18]. HybHPs are located at the periportal region of normal liver and are efficient in liver repair when non-centrilobular hepatocytes are damaged. Thus, it appears that liver injury triggers several regenerative responses depending on the size and the proliferative capacity of the remaining liver tissue[19].

The mechanisms regulating liver cell proliferation and differentiation are highly controlled to achieve accurate tissue growth and development, and deregulation of the signaling pathways involved in liver cell differentiation can impair regeneration and trigger the development of tumors with hepatocellular and cholangio-cellular differentiation features[20,21]. Thus, a better understanding of the processes that control liver cell differentiation in physiological and pathological conditions may contribute to the identification of druggable targets and represent a potential therapeutic approach for the treatment of liver diseases.

Transcription factor EB (TFEB) belongs to the MiT-TFE family of transcription factors, which also includes MITF, TFE3, and TFEC[22]. In the last decade, several studies have explored the role of TFEB in physiological settings and in response to environmental cues. TFEB was first described as essential for placental vascularization[23]. Subsequently, TFEB has been implicated in the expression of lysosomal and autophagic genes[24,25]. TFEB is phosphorylated by the mTOR kinase and participates in a lysosomal signaling mechanism that is essential for nutrient sensing and maintenance of cellular homeostasis and energy metabolism[26–28]. In addition, TFEB has an important role in the regulation of body metabolism in liver and muscle and in the adaptation to environmental cues (e.g. fasting, high-fat diet, exercise)[29,30]. Furthermore, overexpression of MiT-TFE genes has been implicated in the pathogenesis of a variety of tumors,

including renal cell carcinoma, melanoma, and pancreatic cancer[31–34], suggesting their involvement in cell differentiation and proliferation. Consistently, it has been reported that TFEB plays a role in osteoblast[35] and myeloid[36] differentiation by modulating the autophagy–lysosomal pathway. Moreover, TFE3, another member of the MiT-TFE family, was shown to be involved in stem-cell commitment by enabling ESCs to withstand differentiation condition[37,38]. However, a role of TFEB in cell fate determination and liver cancer has not been investigated so far.

Here we show that TFEB plays a critical role in controlling cell fate and proliferation in the mammalian liver during embryogenesis and repair. Indeed, we find that TFEB is highly expressed in LPCs and BECs and that genetic manipulation of TFEB expression in the liver alters cell lineage specification in both developing and injured adult liver. Our data identify *Sox9* as one important downstream target of TFEB in hepatic cell differentiation, highlighting the importance of this pathway in liver development, regeneration, and cancer.

## Results

**TFEB is highly expressed in the biliary compartment**. To investigate whether TFEB plays a role in liver development, we examined its expression during liver specification. We used a mouse line in which the endogenous *Tcfeb* allele is disrupted by homologous recombination at exons 4 and 5 through the insertion of the β-galactosidase coding sequence[26] (Supplementary Fig. 1a). Since *Tcfeb*KO mice are embryonic lethal[23,26], heterozygous embryos (*Tcfeb*LacZ/+) were collected at several embryonic stages to analyze *Tcfeb* promoter activity. *Tcfeb* expression was detected in the liver starting at embryonic stage E14.5 and increases overtime, just prior to the developmental differentiation of HBs into hepatocytes or BECs[39], while no expression was detected at E12.5 (Supplementary Fig. 1b). Consistent with these results, the expression of the endogenous *Tcfeb* mRNA and protein expression levels in fetal and neonatal livers showed a gradual increase during liver growth (Supplementary Fig. 1c, d). Immunostaining analysis revealed a dynamic expression pattern of TFEB during liver development. At E15.5 TFEB was diffuse in the entire parenchyma (Fig. 1a). At postnatal stage P0 and adult stage, a stronger expression was detected in the portal vein endothelium (Fig. 1a, b), while pericentral hepatocytes (adjacent to the central vein) showed low and diffuse TFEB expression (Fig. 1b). Indeed, TFEB was mainly detected in HNF4α−/SOX9+ ductal plate cells and mature ducts (Fig. 1b, c). X-Galactosidase staining in *Tcfeb*LacZ/+ mice showed the same pattern of expression (Supplementary Fig. 1e). Together, our results show that TFEB expression is highly enriched in progenitor/ductal cells, while mature hepatocytes display lower TFEB levels, suggesting a possible role for TFEB in liver cell fate specification.

**TFEB influences liver cell differentiation in vitro**. To investigate whether TFEB plays a role in liver cell commitment, we generated CRISPR/Cas9 *Tcfeb*KO (TFEBKO) (Supplementary Fig. 2a) and TFEB-overexpressing (TFEBOE) mouse HBs. To generate TFEBOE HBs, we isolated HBs from *Tcfeb* conditional overexpressing mice that carry *Tcfeb*-3xFlagfs/fs under the control of a strong CMV early enhancer/chicken beta-actin (CAG) promoter (Supplementary Fig. 2b)[25,32]. To induce *Tcfeb* overexpression, we infected HBs with an HDAd-BOS-CRE virus. HBs are bi-potent cells that can differentiate into hepatocytes or cholangiocytes in culture when plated on uncoated or matrigel coated plates, respectively[40]. Consistently, control HBs differentiated into hepatocytes forming hepatocyte clusters (Fig. 2a) and showed induction of the expression of hepatocyte-specific genes such as *Alb*, *Hnf4α*, *AldoB*, and *Otc*, with concomitant reduction of the

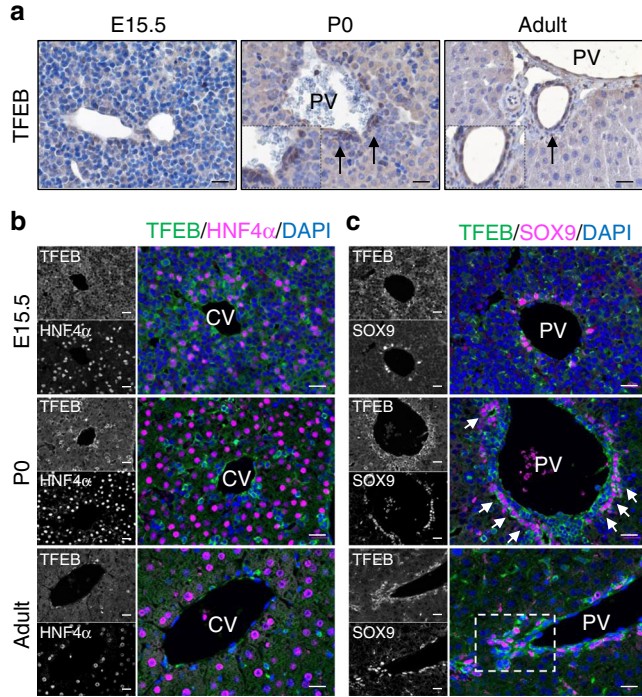

**Fig. 1 TFEB expression is enriched in ductal/progenitor cells.**
**a** Immunohistochemistry analysis of TFEB in wild-type (WT) liver at the indicated stages showing prominent signaling in bile ductules. Arrows indicate ductal cells. **b, c** Representative immunofluorescence stains for TFEB/HNF4α (**b**) and TFEB/SOX9 (**c**) showing TFEB levels in the central (CV) and in the portal (PV) vein area. Scale bar 20 μm. Micrographs are representative of five mice of each stage.

expression of the precursor markers *Sox9*, *Sox4*, *Afp*, and *Cd24* (Fig. 2b). Notably, HBs lacking TFEB showed significantly increased expression of the hepatocyte-specific markers and reduced *Sox9* levels compared to controls, with no differences in the expression of the precursor markers *Afp*, *Sox4*, and *Cd24* (Fig. 2b), suggesting that TFEB loss-of-function preferentially induces the hepatocyte differentiation program. On the contrary, TFEB-overexpressing HBs did not completely differentiate into hepatocytes, as demonstrated by the smaller size of the hepatocyte-like aggregates (Fig. 2a), lower expression levels of hepatocyte-specific genes, and higher levels of the precursor-specific markers (Fig. 2b), suggesting that TFEB overexpression prevents the hepatic differentiation of HBs, while maintaining precursor features. These results were confirmed by immunoblot and immunofluorescence analysis on HBs 3 days after hepatocytic differentiation (Fig. 2c, d).

To confirm the defective differentiation program, we examined the effect of TFEB overexpression and depletion on the transcriptome of HBs 3 days after differentiation toward the hepatocytic lineage. Transcriptional profiles of TFEB^KO and TFEB^OE HBs confirmed the altered expression of hepatocyte- and progenitor/cholangiocyte-specific genes (Fig. 2e). KEGG analysis on the differentially expressed genes showed upregulation of hepatocyte-specific pathways, such as drug metabolism—cytochrome P450, fat digestion and absorption, and cholesterol metabolism in TFEB^KO HBs that were instead downregulated in TFEB^OE HBs (Supplementary Tables 1 and 2). Moreover, gene set enrichment analysis (GSEA) demonstrated that HNF4α targets (a master transcriptional regulator of hepatocyte differentiation) are enriched among upregulated genes in TFEB^KO HBs and show a reduced expression, despite not significant, in TFEB^OE HBs (Supplementary Fig. 2c). We also examined the distribution of

HBs in the three phases of the cell cycle (G1 vs S vs G2/M). TFEB^KO HBs showed significant increase in the percentage of S phase cells compared to controls, suggesting that TFEB depletion induces S phase arrest in HBs (Fig. 2f). Interestingly, TFEB overexpression in HBs resulted in increased percentage of cells in the G2/M phase compared with control cells with a concomitant reduction in the S phase (Fig. 2f), suggesting increased proliferation.

In contrast to the complete inhibition of hepatocyte specification, neither TFEB depletion nor overexpression impaired biliary differentiation of HBs. Indeed, TFEB^KO and TFEB^OE cells supported tubule formation (Supplementary Fig. 2d) and expressed biliary genes (i.e. *Hif1α, Hnf6, Ggt1*) after cholangiocytic differentiation despite the reduced or increased expression levels of both *Sox9* and *Afp*, respectively (Supplementary Fig. 2e).

Together these results indicate that TFEB plays a role in the lineage commitment of liver precursor cells.

**TFEB^OE induces a progenitor/biliary phenotype in vivo.** To test whether TFEB overexpression alters cell differentiation from a progenitor state and impairs homeostasis of mature hepatocytes in vivo, we crossed the *Tcfeb*-3xFlag^fs/fs mouse line (Supplementary Fig. 2b) with a transgenic line carrying *Alb*-Cre recombinase to obtain *Tcfeb*-3xFlag^fs/fs;*Alb*-Cre mice (hereafter referred to as Tg). *Albumin* is expressed by the bipotential HB progenitors, thus enabling us to investigate the role of TFEB during liver cell specification. Tomato expression and in situ hybridization analysis confirmed TFEB overexpression in hepatocytes and BECs at E18.5, P0, and P9 (Supplementary Fig. 3a, b). HB-specific expression of TFEB was assessed by qPCR analysis on liver extracts of Tg mice showing an approximately 10-fold increase of *Tcfeb* mRNA levels in E18.5, P0, and P9 livers and up to 65-fold in 3-month-old mice (Supplementary Fig. 3c), consistent with the progressive increase in *Alb* mRNA during liver specification[41].

To examine the effects of TFEB overexpression in vivo, we carried out microarray analysis at an early stage (P9) showing a total of 8400 differentially expressed genes. KEGG analysis revealed that several upregulated genes are involved in oxidative phosphorylation, proteasome, cell cycle, Hippo signaling pathway and endocytosis, among others, while downregulated genes are mostly involved in hepatocyte-specific pathways, such as amino acid, cholesterol, and lipid metabolism, drug metabolism (P450), PPAR signaling pathway (Fig. 3a, Supplementary Table 3). Consistent with in vitro data, gene expression profile of Tg livers demonstrated a reduction in the expression of hepatocyte-specific genes and an increase in progenitor/cholangiocyte related genes compared with control livers (Fig. 3b, c, Supplementary Table 4). Immunoblotting analysis confirmed the reduction of HNF4α and the increase of SOX9 protein levels in liver extracts from Tg mice at P9 (Fig. 3d). Similar results were obtained in primary hepatocytes isolated from 21-day-old Tg and control mice (Supplementary Fig. 4a, b). Interestingly, while no CK19^+ cells were detected in control hepatocytes, we found HNF4α^+/CK19^+ Tg cells, suggesting a hybrid phenotype of Tg hepatocytes (Supplementary Fig. 4c). Indeed, immunostaining for the cholangiocyte marker CK19 in Tg livers from P0 mice showed CK19^+ cells with hepatocyte-like morphology positive for the hepatocyte marker HNF4α mainly in the periportal area (Fig. 3e, Supplementary Fig. 5a). The concomitant positive signals for both a cholangiocyte and a hepatocyte marker suggest defective differentiation. At P9, Tg mice exhibited dilated bile ducts and ectopic tubules throughout the lobule (Fig. 3e, Supplementary Fig. 5a). Immunofluorescence analysis showed that most of the ductal CK19^+ cells were positive for SOX9

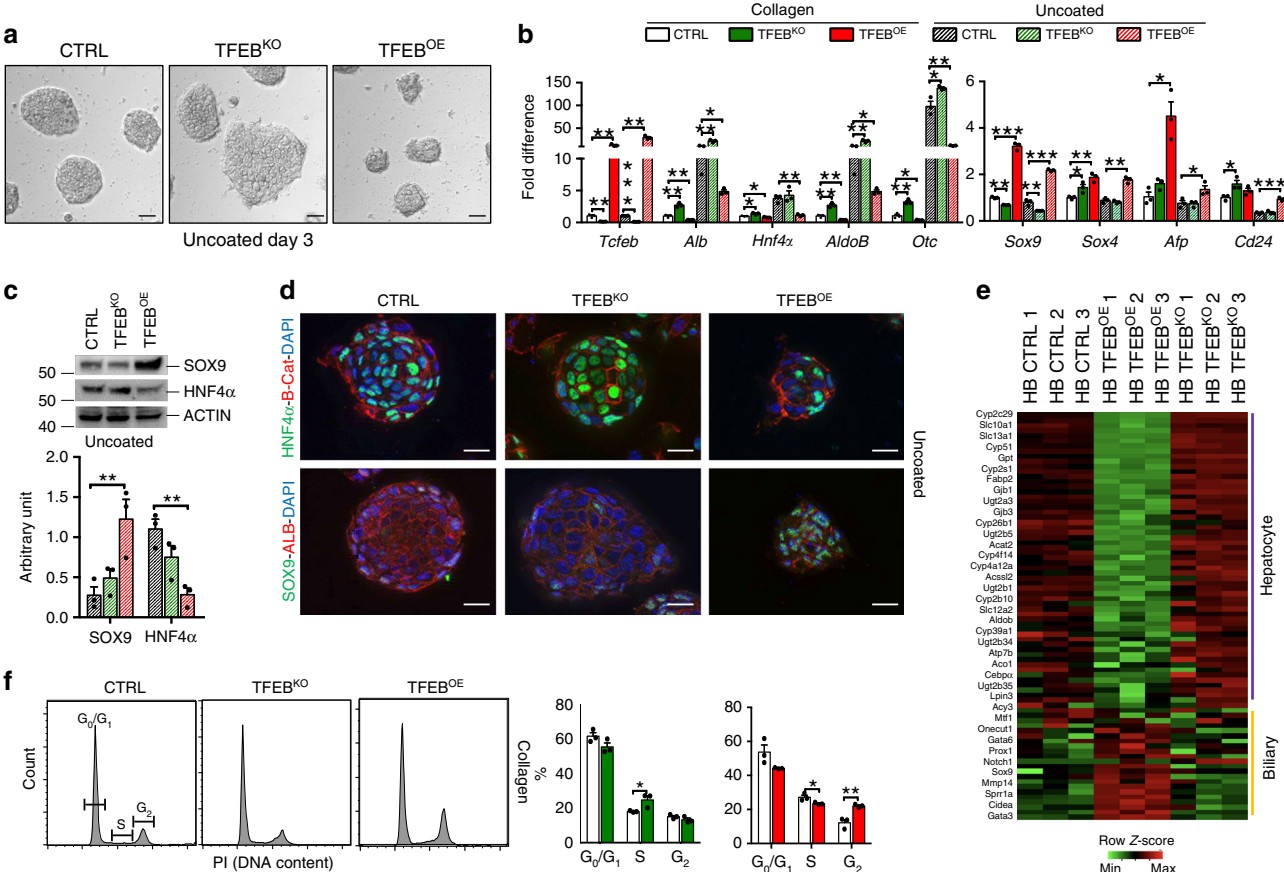

**Fig. 2 TFEB influences hepatoblast differentiation in vitro. a** Hepatocyte sphere formation of HBs of the indicated genotypes 3 days after differentiation. Scale bar 20 μm. Micrographs are representative of three independent experiments. **b** mRNA levels of the indicated genes were quantified by quantitative RT-PCR of total RNA isolated from control (CTRL), TFEB-overexpressing (TFEB[OE]), and TFEB depleted (TFEB[KO]) HBs undifferentiated (collagen-coated plates) or after 5 days of hepatocyte differentiation (uncoated plates). Values are indicated as mean ± SEM of $n = 3$ biological replicates and expressed as fold difference compared with CTRL undifferentiated HBs. **c** Immunoblotting analysis and relative quantification of the precursor/cholangiocyte marker SOX9 and the hepatocyte marker HNF4α in TFEB[KO] and TFEB[OE] HBs after 5 days of hepatocytic differentiation ($n = 3$ biological replicates). **d** Immunostaining for the indicated markers on TFEB[KO] and TFEB[OE] HBs 5 days after hepatocytic differentiation. Scale bar 5 μm. Micrographs are representative of three independent experiments. **e** Gene expression profiling of hepatocyte and biliary genes of differentiated HBs of the indicated genotypes. **f** Flow cytometry analysis of the cell cycle distribution of undifferentiated CTRL, TFEB[KO] and TFEB[OE] HBs and relative quantification ($n = 3$ biological replicates). Source data are provided in the Supplementary Fig. 10. Data are represented as mean ± SEM. $*p \leq 0.05$, $**p \leq 0.01$, $***p \leq 0.001$ two-tailed Student's $t$-test. Source data are provided as a Source Data file.

(Supplementary Fig. 5b), as expected, but other cells scattered in the entire parenchyma were HNF4α+/SOX9+/CK19− in P0 and P9 livers (Fig. 3e, Supplementary Fig 6a), suggesting that they were hepatocytes with bi-phenotypic features. In situ hybridization analysis confirmed that SOX9 was expressed in the entire liver in Tg mice compared to controls (Supplementary Fig. 6b).

Together, these observations suggested that TFEB overexpression directs cell fate of HBs towards a progenitor/cholangiocyte lineage.

**Sox9 is a direct target of TFEB.** Our transcriptomic analysis in TFEB[OE] HBs and livers showed enrichment in several progenitor-specific genes, including *Sox9*. *Sox9* overexpression is known to be sufficient to induce biliary genes and suppress differentiation into the hepatocyte lineage[42]. Therefore, we investigated whether *Sox9* is a direct transcriptional target of TFEB and a mediator of TFEB effects on liver cell differentiation. We analyzed the promoter region of *Sox9* gene and identified five putative TFEB target sites (i.e. the CLEAR sites)[24] that were validated by chromatin immunoprecipitation (ChIP) analysis.

Compared with a control sequence, the sequences closest to the *Sox9* transcriptional start site (TSS) were significantly enriched in liver samples from Tg mice (Fig. 4a). The region containing CLEAR sites closest to the TSS was cloned into a luciferase reporter plasmid to evaluate its responsiveness to TFEB in HBs. Overexpression of TFEB increased luciferase activity in HBs, whereas deletion of the CLEAR sites failed to induce transactivation (Fig. 4b). Taken together these data indicate that TFEB binds to the *Sox9* promoter, identifying *Sox9* as a direct transcriptional target of TFEB.

To further validate our findings, we performed genetic interaction studies by crossing *Tcfeb*-3xFlag[fs/fs];*Alb*-Cre with *Sox9*[f/f] mice to generate double transgenic *Tcfeb*-3xFlag[fs/fs];*Alb*-Cre;*Sox9*[f/+] mice (hereafter Tg;*Sox9*[f/+]). The expression of *Sox9* was significantly reduced in livers from Tg;*Sox9*[f/+]compared with Tg mice (Fig. 4c, d). qPCR analysis on livers from Tg;*Sox9*[f/+] at P9 showed higher expression levels of hepatocyte-specific markers (e.g. *Alb*, *AldoB*, and *Otc*) and a reduction of cholangiocyte-markers (e.g. *Krt7* and *Krt19*) compared with Tg mice (Fig. 4c). Interestingly, immunostaining analysis showed that *Sox9* knockdown resulted in reduced number of CK19+ cells

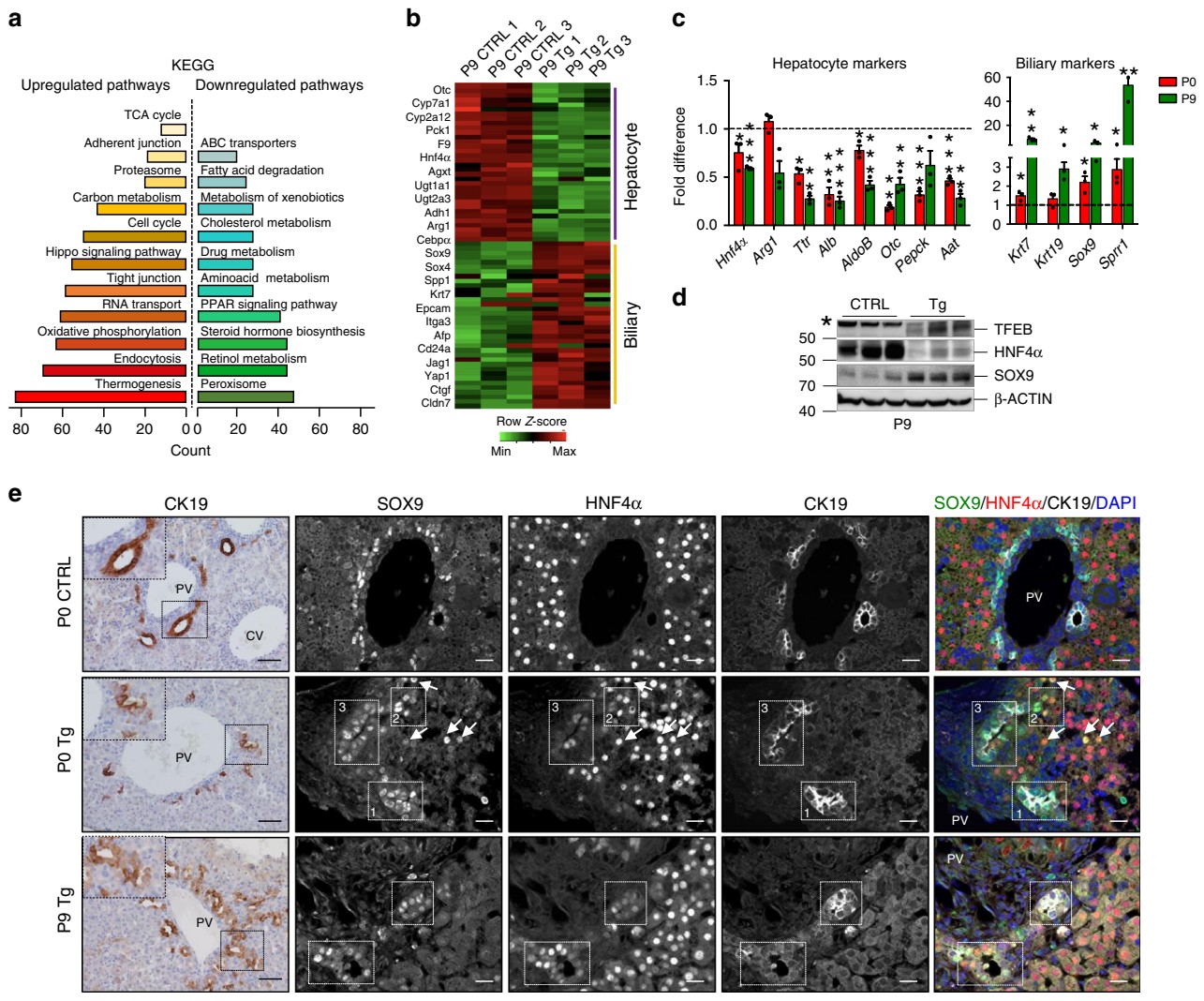

**Fig. 3 TFEB overexpression inhibits hepatocyte differentiation in vivo. a** KEGG pathways enriched in upregulated and downregulated genes of P9 liver overexpressing TFEB compared to CTRL. **b** Heat map showing the relative fold change of hepatocyte and biliary genes. **c** RT-PCR analysis of hepatocyte- and cholangiocyte-specific markers in liver isolated from P0 and P9 mice ($n = 3$ biological replicates). The dashed line represents gene expression levels in CTRL liver. Data are represented as mean ± SEM. *$p \leq 0.05$, **$p \leq 0.01$ two-tailed Student's $t$-test compared to age-matched CTRL. **d** Immunoblotting analysis of TFEB, HNF4α, and SOX9 in liver extracts from mice at P9. Actin was used as a loading control. **e** Left panel: Representative images of liver sections from P0 and P9 mice stained for CK19. Scale bar 50 μm. Right panels: triple immunostaining for HNF4α (red), SOX9 (green), and CK19 (white) in liver sections from P0 and P9 mice of the indicated genotype showing the hybrid characteristics of hepatocytes and cholangiocytes in Tg mice. Note the cells expressing SOX9 and CK19 (box1), SOX9 and HNF4α (box2) or SOX9/HNF4α/CK19 (box3). Scale bar 20 μm. Source data are provided as a Source Data file.

compared to Tg liver (Fig. 4e). These data suggest that SOX9 mediates, at least partially, the effects of TFEB on the determination of liver cell fate.

**TFEB depletion impairs LPC expansion upon liver injury**. We then investigated the effects of TFEB deletion on liver cell differentiation in a previously described TFEB liver-specific conditional KO mouse line *Alb*-Cre;*Tcfeb*f/f (*Tcfeb*LiKO)[26]. While immunofluorescence analysis did not reveal any significant cell differentiation defect in TFEB-depleted livers compared to age-matched controls at P0 and 1 month (Supplementary Fig. 7), DDC-induced liver damage (Fig. 5a) in *Tcfeb*LiKO mice resulted in a reduction in the expression of *Sox9* and *Krt19* during the recovery phase (Fig. 5b), which was confirmed by immunoblot analysis and immunostaining for SOX9 at 14 days after recovery (Fig. 5c, d). These mice also showed a reduced ductular reaction as measured by CK19 immunostaining (Fig. 5d). Interestingly,

while control livers showed hybrid hepatocytes positive for both HNF4α and SOX9 around the portal vein as a consequence of liver damage, as previously shown[43], *Tcfeb*LiKO mice showed SOX9+ cells only in the portal area (Fig. 5d, e) and a strong reduction of proliferating SOX9+ and CK19+ cells, indicating impaired LPC activation (Fig. 5f, g). No main differences were observed in weight recovery after injury in the two genotypes (Supplementary Fig. 8a), while an increase in serum markers of liver damage (e.g. bilirubin and AST) was detected in *Tcfeb*LiKO mice compared to control mice (Supplementary Fig. 8b). These data suggest that liver cell differentiation after injury requires TFEB induction. Consistently, we found that TFEB translocates from the cytoplasm to the nucleus 7 days after recovery from DDC-induced liver injury (Supplementary Fig. 8c), a time point during which we detected a reduction in mTOR activity, as measured by detecting the phosphorylation of the mTOR substrate 4EBP1 (Supplementary Fig. 8d).

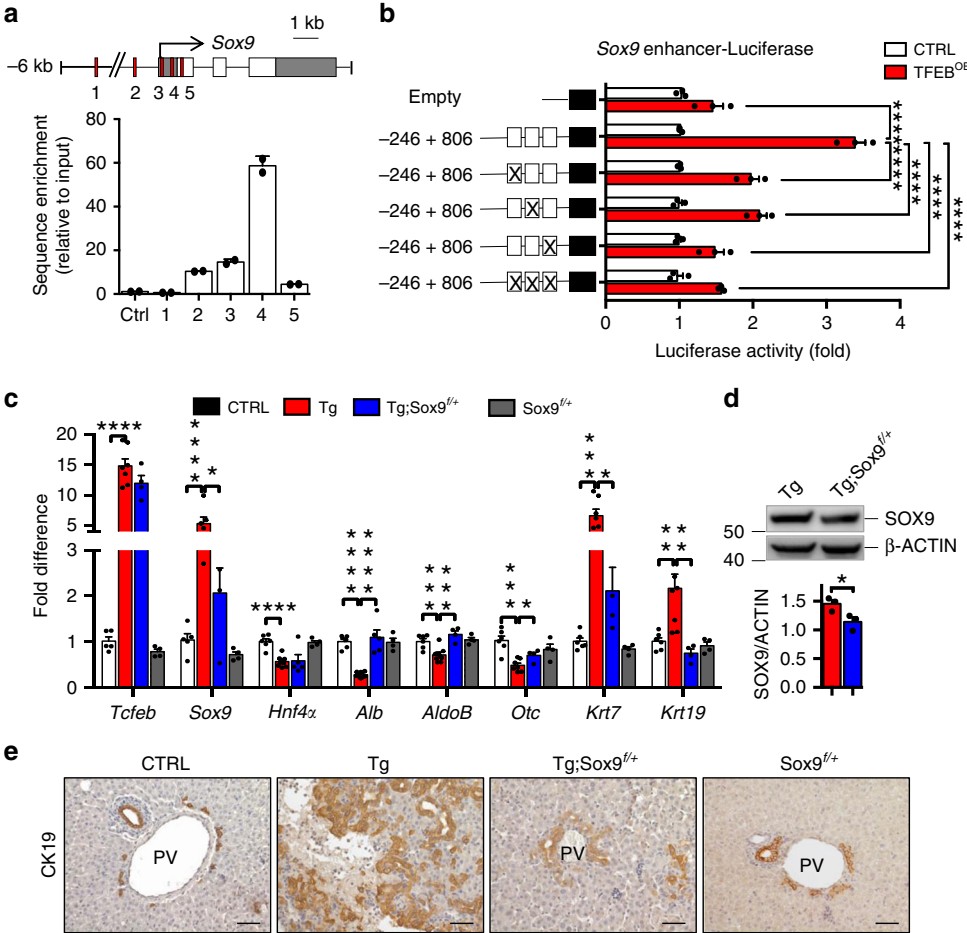

**Fig. 4 TFEB directly controls *Sox9* expression. a** ChIP analysis of TFEB binding to the *Sox9* promoter in Tg livers. CLEAR sites in the promoter region of *Sox9* are indicated by red squares. Bar graphs show the amount of immuno-precipitated DNA as detected by RT-PCR analysis. Values are representative of two independent experiments. Data were normalized to the input and plotted as relative enrichment over the control sequence. **b** Luciferase expression upon TFEB overexpression was measured in HBs infected with HDAd-LacZ (CTRL) or HDAd-CRE expressing viruses (TFEB^OE) for 24 h and transfected with wild type or mutated *Sox9*-promoter luciferase reporter plasmids ($n = 3$ biological replicates). Deletion of the CLEAR sites resulted in reduced luciferase transactivation. Data are represented as mean ± SEM. ****$p \leq 0.0001$ two-way ANOVA. **c** RT-PCR expression analysis of hepatocyte- and cholangiocyte-specific markers in livers isolated from P9 mice for the indicated genotypes ($n = 6$ CTRL, $n = 8$ Tg, $n = 4$ Tg; Sox9^f/+, $n = 4$ Sox9^f/+). Data are represented as mean ± SEM. *$p \leq 0.05$, **$p \leq 0.01$, ***$p \leq 0.001$, ****$p \leq 0.0001$ two-tailed Student's *t*-test. **d** Representative immunoblot of SOX9 in liver of the indicated genotypes and relative quantification ($n = 3$ biological replicates). **e** CK19 immunostaining of liver sections from P9 mice of the indicated genotypes. Scale bar 50 μm. Source data are provided as a Source Data file.

**TFEB^OE in regenerating adult liver subverts cell identity**. The effects of TFEB overexpression in the progenitor/cholangiocyte compartment were investigated by crossing the *Tcfeb*-3xFlag^fs/fs; *R26R^LSL tdTomato* mice with an inducible *Krt19*^CreERT mouse line that labels the biliary epithelium with a 40% recombination efficiency (Fig. 6a, b)[10]. Four-weeks after tamoxifen injection, tdTom expression was observed only in biliary ducts and in small periportal ductules (Fig. 6a). Based on the results obtained in *Alb*-Cre mice, we hypothesized that TFEB overexpression would expand the CK19^+ population in the periportal area by generating a ductal reaction. Indeed, we observed an increase in CK19^+/tdTom^+ cells in Tg mice compared to controls (Fig. 6b), hyperplasia of the bile ducts, and a population of cells growing within the ductal epithelium and forming multilayered structures (Fig. 6a). In addition, some of the CK19^+/tdTom^+ cells appeared with a rounded morphology, compared to the cuboidal biliary epithelium observed in control mice, with a migrating phenotype that mimics a ductular reaction (Fig. 6a). This phenotype strongly resembles the one observed in YAP-overexpressing biliary cells[44].

Subsequently, to stimulate a progenitor/biliary-derived liver regenerative response, we induced chronic liver damage by feeding mice with a 0.1% 3,5-diethoxycarbonyl-1,4-dihydrocolli-dine (DDC)-containing diet for 3 weeks and then switch to a normal diet for 14 days, a protocol causing transient LPC activation (Fig. 6c)[5]. Interestingly, while control mice showed tdTom^+ cholangiocytes in the portal area and the expected ductular reaction as shown by tdTom, CK19 and SOX9 immunostaining (Fig. 6d), most of the CK19^+ and SOX9^+ cholangiocytes were tdTom^+ in Tg mice, suggesting increased proliferation of the tdTom^+ cells upon TFEB overexpression, as also confirmed by BrdU analysis (Fig. 6d). Notably, these mice recapitulate the same phenotype observed in the *Alb*-Cre;*Tcfeb*-3xFlag^fs/fs mice.

To overexpress TFEB specifically in the hepatocytes, we injected *Tcfeb*-3xFlag^fs/fs;*R26R^LSL tdTomato* mice with a CRE-expressing adeno-associated virus (AAV8-TBG-CRE). As expected, AAV8 injection induced CRE expression in 99% of the hepatocytes and no expression in the biliary cells as demonstrated by tdTom staining (Fig. 7a)[5]. These mice showed

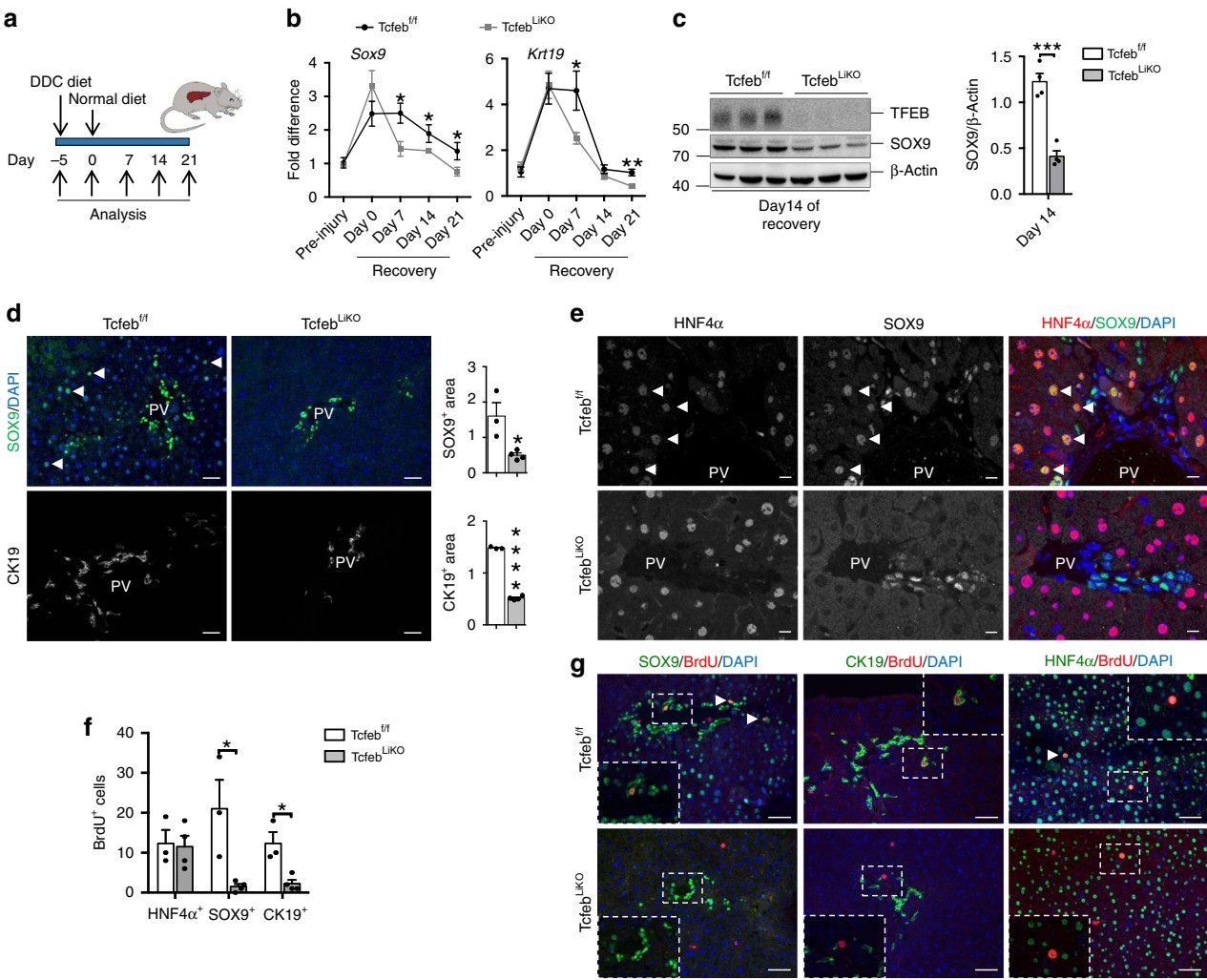

**Fig. 5 TFEB depletion impairs ductular reaction and cell proliferation after liver injury. a** Experimental strategy for liver injury protocol for TFEB[LiKO] mice. **b**, **c** Expression levels of *Sox9* and *Krt19* mRNA (pre-injury $n = 3$, day 0–7–14–21 $n = 5$ biological replicates per genotype) at different time points (**b**) and immunoblotting with relative quantification ($n = 4$ biological replicates) (**c**) of livers isolated from TFEB[f/f] and TFEB[LiKO] mice 14 days after discontinuation of DDC food. In **b**, values are expressed as fold difference relative to TFEB[f/f] mice. Data are represented as mean ± SEM. \*$p \leq 0.05$, \*\*$p \leq 0.01$, \*\*\*$p \leq 0.001$ two-tailed Student's *t*-test. **d** SOX9 and CK19 immunofluorescence with relative quantifications (TFEB[f/f] $n = 3$, TFEB[LiKO] $n = 4$ biological replicates/$n = 5$ different images/sample) of liver isolated from TFEB[f/f] and TFEB[LiKO] mice 14 days after recovery and relative quantifications. Scale bar 50 μm. **e** HNF4α/ SOX9 dual staining of liver sections from TFEB[f/f] and TFEB[LiKO] mice. Scale bar 5 μm. **f**, **g** Quantification of BrdU+ cells ($n = 3$ biological replicates/$n = 5$ different images/sample) (**f**) and representative images (**g**). Scale bar 50 μm. Data are represented as mean ± SEM. \*$p \leq 0.05$ two-tailed Student's *t*-test. Source data are provided as a Source Data file.

20-fold increase of *Tcfeb* expression at 4 weeks after injection (Fig. 7b), but no alteration in cell identity compared to controls (Fig. 7c). These results suggest that TFEB overexpression in differentiated adult hepatocytes is not sufficient to induce their conversion to cholangiocytes.

To evaluate whether TFEB overexpression in hepatocytes alters cell identity upon liver injury, we fed *Tcfeb*-3xFlag[fs/fs]; *R26R*[LSL]*tdTomato* mice injected with AAV8-TBG-CTRL or AAV8-TBG-CRE with the DDC-containing diet. We first used a protocol of acute injury by feeding mice a DDC-containing diet for 5 days and switching to a normal diet for 14 days (Fig. 7d). Interestingly, CRE-injected mice showed tdTom+ hepatocytes positive for SOX9 and CK19, strongly suggesting a conversion of TFEB[OE] cells in progenitor/cholangiocyte-like cells upon liver damage (Fig. 7e). Notably, the chronic exposure to DDC-containing food for 3 weeks following by 14 days of recovery resulted in liver tumors in CRE-injected mice (Fig. 7f, g). Histological analysis showed that most of the CK19+/SOX9+ cells

were tdTom+, and that some tdTom+ hepatocytes also expressed SOX9 (Fig. 7h), confirming that TFEB overexpression in hepatocytes leads to trans-differentiation in progenitor/cholangiocytes in conditions of liver damage when cells are more prone to proliferation and trans-differentiation.

Together our data strongly confirm that TFEB overexpression in progenitors/cholangiocytes is able to influence proliferation and differentiation.

**Cholangiocarcinoma-like phenotype in TFEB[OE] mice.** We then analyzed the phenotypic consequence of the altered cell differentiation associated with TFEB overexpression in the liver. Tg mice exhibited significant hepatocellular derangement as shown by an increase of liver damage markers, as well as a significant increase in both total bilirubin and direct bilirubin at 15 days and 3 months (Supplementary Table 5) consistent with cholestasis. Serum bile acid and cholesterol levels were also increased in Tg

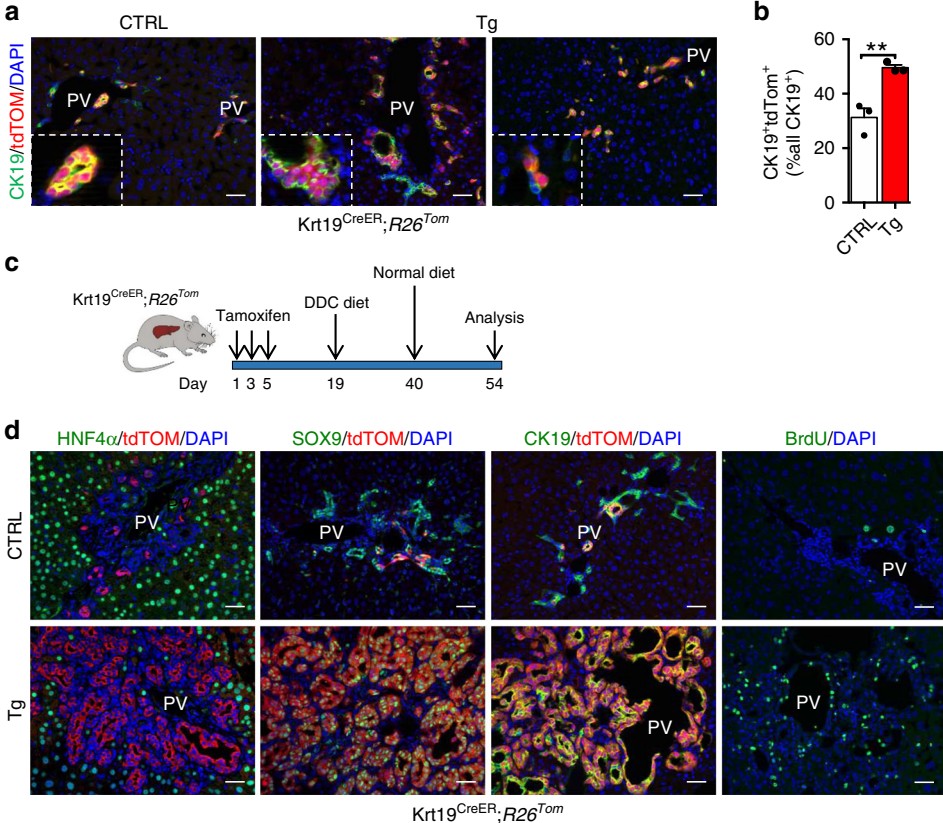

**Fig. 6 TFEB overexpression in progenitor/cholangiocytes gives rise to ectopic ductal structures. a** Representative image of CK19/tdTom dual immunofluorescence of CTRL and TFEB$^{OE}$ liver 4-weeks after tamoxifen injection. Scale bar 50 μm. **b** Quantification of CK19$^+$ biliary epithelial cells that were tdTom$^+$ ($n = 3$ biological replicates/$n = 5$ different images/sample). Data are represented as mean ± SEM. **$p \leq 0.01$ two-tailed Student's $t$-test. **c** Experimental strategy to lineage trace biliary epithelial cells on a background of liver injury. **d** tdTom/HNF4α, tdTom/SOX9, and tdTom/CK19 immunofluorescent images at 14 days after DDC diet. Scale bar 50 μm. Source data are provided as a Source Data file.

mice compared to controls (Supplementary Table 5). Analysis of three-dimensional structure of the biliary system in adult liver, performed by retrograde injection of ink in the biliary tree in 2-month-old mice, confirmed a defect in bile duct development (Fig. 8a). Indeed, we observed reduced density of branches arising from major branches and abnormal major branches in appearance consistent with partial obstruction of bile flow, correlating with increased bilirubin and ALT levels. Furthermore, Tg mice showed a progressive increase in liver mass relative to total body mass with a 2-, 4-, and 10-fold increase at 2 weeks, 2 months, and 5 months of age, respectively (Fig. 8b, c). At 3 months, TFEB overexpression was detected in the entire parenchyma and in particular in the dilated ducts as demonstrated by immunohistochemistry and in situ hybridization analysis (Fig. 8d). Livers from Tg mice displayed variable degrees of hyperplasia of bile ducts, which were increased in size and number, and an altered morphology of epithelial cells. In addition, they showed multifocal biliary cystic hyperplasia with cysts lined by flattened epithelium at 3 months of age (Fig. 8e). Densely packed cholangiocytes, as revealed by CK19 immunostaining, associated with increased fibrosis revealed by Sirius red staining, were also detected suggesting the presence of a neoplasms of the bile ducts. All of these phenotypic features are consistent with the presence of a cholangiocarcinoma (CCA)-like phenotype. Livers from 6-month-old Tg mice showed multiple cysts replacing most of the hepatic parenchyma. At this age mice began to die or needed to be euthanized due to poor health status. Increased liver mass in Tg mice was associated with a higher proliferation index compared to control mice, as shown by increased Ki67 staining

(Fig. 8e). A similar, albeit milder and with later onset, phenotype was observed in a different transgenic mouse line in which *Tcfeb* is overexpressed at lower levels (2.5-fold increase) suggesting a dose- and time-dependent effect of TFEB overexpression (Supplementary Fig. 9a–d).

These data are consistent with a pro-cholangiocytic function of TFEB and suggest that TFEB induction may be involved in the pathogenesis of cholangiocarcinoma.

## Discussion

Recent studies have identified an important role for MiT-TFE transcription factors (TFs) in the regulation of basic cellular processes, such as lysosome, autophagosome, and melanosome biogenesis[24,25,45,46]. These TFs control organismal adaptation to environmental cues, such as nutrient availability, physical exercise, and infections[29,30,34,47–51]. While some studies reported the involvement of MiT-TFE genes in cell differentiation, the precise role that these TFs play in cell specification and in embryonic development has remained elusive. TFEB full KO mice die at E10 due to a placental vascularization defect, indicating that embryonic development requires TFEB-mediated transcriptional control[23]. In addition, previous studies showed that MITF and TFEB are involved in the differentiation processes of melanocytes and osteoclasts, respectively[52]. A recent study also reported a role for TFEB in the activation and response to growth factor in quiescent neuronal stem cells (qNSCs)[53]. Furthermore, TFE3, another member of the MiT family, was shown to be involved in stem-cell commitment by enabling ESCs to withstand

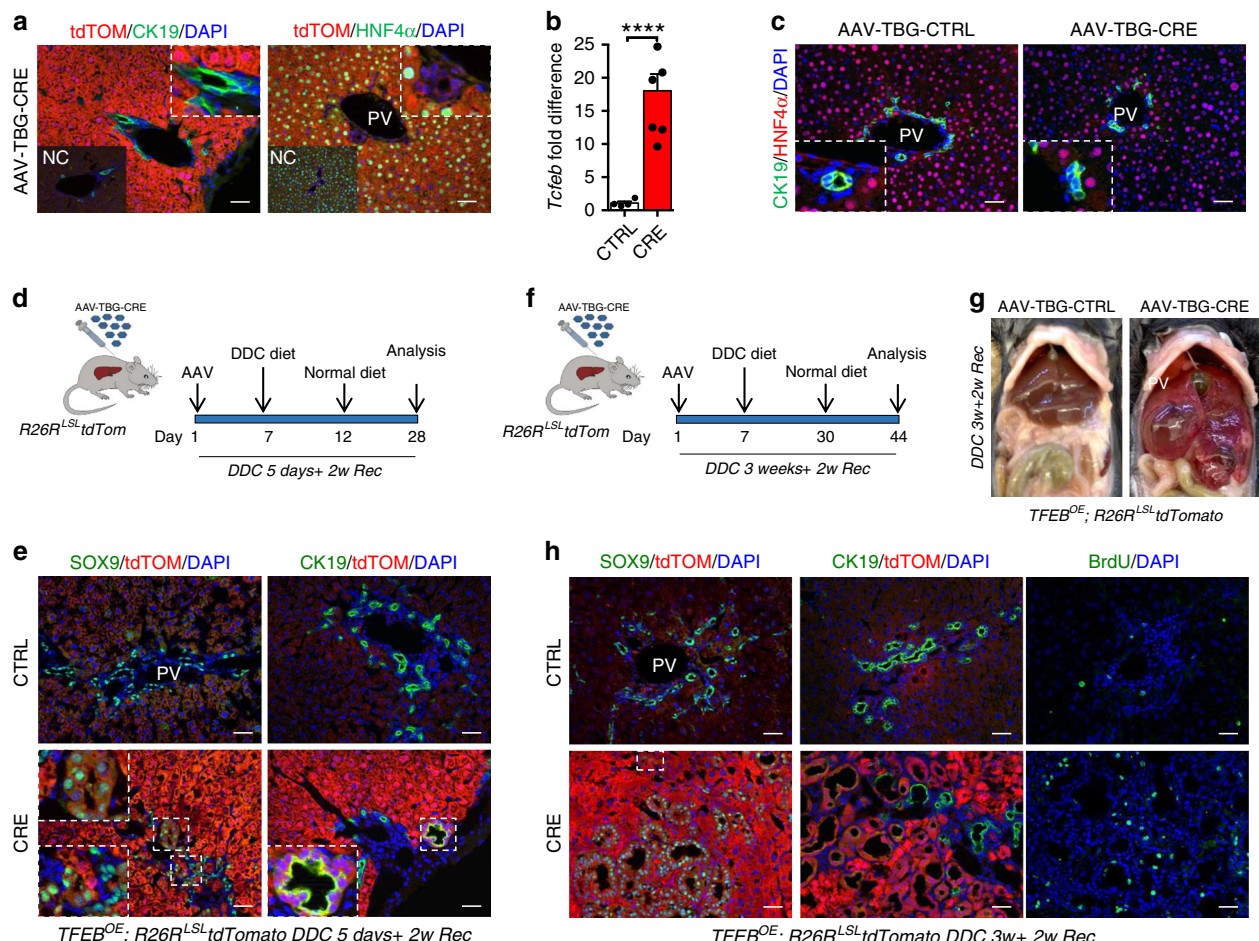

**Fig. 7 TFEB-overexpressing hepatocytes dedifferentiate in progenitor/cholangiocyte-like cells upon liver injury. a** tdTom/CK19 and tdTom/HNF4α immunofluorescence in AAV8-TBG-CRE-treated livers 4 weeks after injection. CRE-treated HNF4α+ hepatocytes are tdTom+ in contrast to CK19+ ductal cells located in the portal tract (PV). Insets: images of AAV-CTRL injected liver indicated as negative control (NC). Scale bar 50 μm. **b** Transcript levels of *Tcfeb* in livers isolated from AAV8-TBG-CRE-treated mice 4 weeks after injection (CTRL n = 4, CRE n = 6 biological replicates). *Tcfeb* mRNA levels were normalized to the CTRL. Data are represented as mean ± SEM. ****p ≤ 0.0001 two-tailed Student's t-test. **c** CK19/HNF4α immunofluorescence analysis in AAV8-TBG-CRE-treated and CTRL liver 4 weeks after injection showing no alteration in cell identity. Scale bar 50 μm. **d** Experimental design of the AAV8 injection followed by acute liver injury protocol. **e** Fourteen days post DDC acute injury hepatocytes dedifferentiate in progenitor/biliary cells. **f** Experimental design of the AAV8 injection followed by chronic liver injury protocol. **g** Gross appearance of AAV-injected livers from day 14 of recovery after DDC prolonged exposure. **h** Immunofluorescence analysis of AAV8-treated livers at day 14 of recovery after DDC diet. Scale bar 50 μm. Source data are provided as a Source Data file.

differentiation condition[37]. In the present study, we reveal an unpredicted role for TFEB in liver cell differentiation during development and in the regenerative response to injury. We found that TFEB expression levels increase overtime during liver specification. Moreover, we observed differences in the levels of TFEB expression between pericentral vs periportal area suggesting that different levels of TFEB could determine different progenitor cell fates. Indeed, our data show that TFEB mainly specify the cholangiocyte lineage. This is also supported by the finding that TFEB overexpression in HBs dictates a progenitor/cholangiocytic fate resulting in a rapid replacement of the entire liver by biliary structures. In this regard, it will be interesting to determine the mechanisms that allow the differential expression of TFEB in specific subsets of cells.

Hepatocyte identity is defined by the expression of a core group of transcription factors primarily driven by the CCAAT/enhancer binding protein alpha (C/EBPα)[54], a key hepatic transcription factor that also controls the expression of genes involved in ammonia detoxification and glucose and lipid homeostasis. Another key regulator of cell identity in the liver is SOX9, a transcription factor

that drives bile duct morphogenesis and is recognized as a specific marker of precursors and biliary cells in the developing liver[55]. Remarkably, C/EBPα and SOX9 form a mutually antagonistic system controlling the hepatocyte versus biliary fate during normal liver homeostasis and regeneration[42]. Our in vitro and in vivo studies indicate that TFEB directly controls the expression of *Sox9* during progenitor/ductal specification and after liver injury. Indeed, our epistatic analysis using *Sox9*-deficient mice provides evidence that TFEB-mediated regulation of SOX9 is required for proper differentiation of bipotential progenitors, although additional mechanisms downstream of TFEB are likely to play a role.

Consistent with a role of TFEB in the determination of liver cell fate, we found that mice overexpressing TFEB in the liver showed increased proliferation of progenitor cells, expansion of immature BECs and inhibition of the hepatocyte lineage. An opposite phenotype was observed in *Tcfeb*-depleted HBs and liver, which appeared more prone to differentiation into the hepatocyte lineage (Fig. 9).

Although liver parenchymal cells turn over slowly, the liver displays high regenerative capacity, capable of restoring 70% tissue

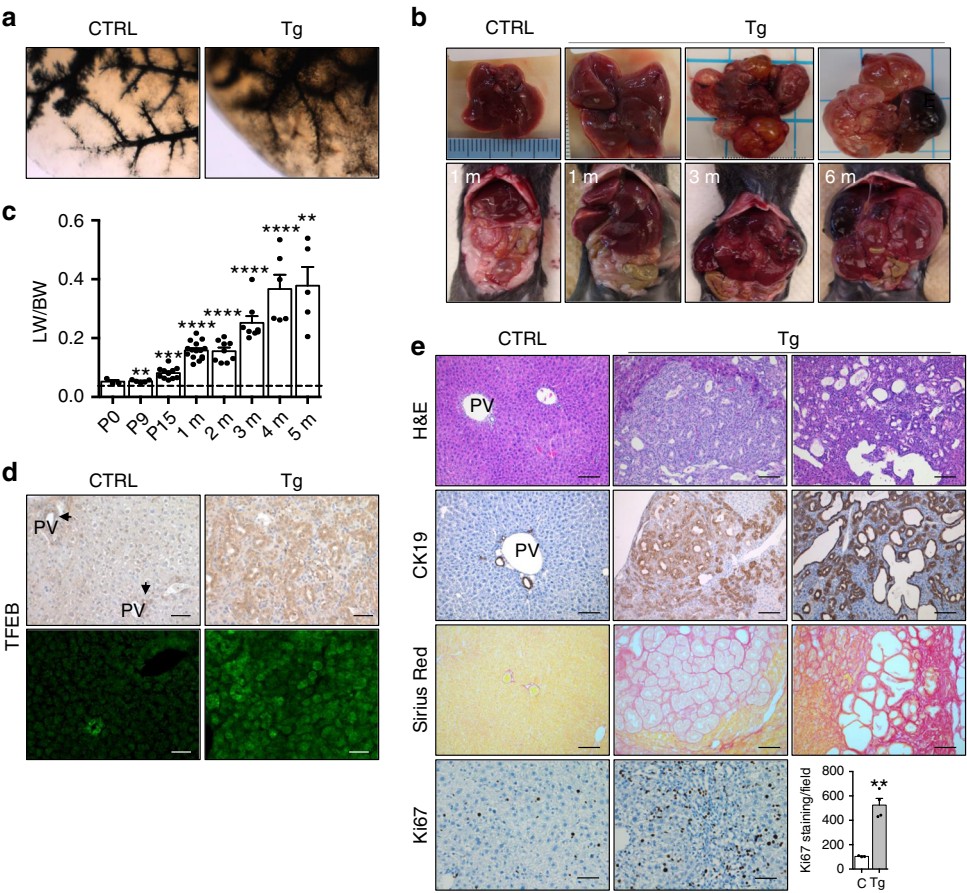

**Fig. 8 Bile duct neoplasm in TFEB-overexpressing liver. a** Retrograde injection of ink revealed abnormal development of the bile duct in Tg liver compared to control. **b** Gross appearance of livers in Tg mice at several ages. **c** Quantification of liver-to-body weight ratio (LW/BW) in Tg mice at the indicated age. ($n = 3$ P0, $n = 5$ P9, $n = 11$ P15, $n = 15$ 1 month, $n = 9$ 2 months, $n = 8$ 3 months, $n = 6$ 4 months, $n = 5$ 5 months). Values were normalized to age-matched control liver as indicated by the dashed line. **d** Immunohistochemistry and in situ hybridization analysis for TFEB in CTRL and Tg liver at 3 months of age. Scale bar 50 μm. **e** Histological characterization of Tg liver phenotype at 3 months of age. Lower panel: Ki67 staining and relative quantification showing increased cell proliferation in Tg liver compared to CTRL (CTRL $n = 3$, Tg $n = 4$ biological replicates/$n = 5$ different images/sample). Data are represented as mean ± SEM. ***$p \leq 0.001$, ****$p \leq 0.0001$ two-tailed Student's $t$-test. Scale bar 100 μm. Source data are provided as a Source Data file.

loss within a few weeks[56]. This ability is vital for the liver to maintain constant mass. The population of liver cell responding to injury may differ depending on the site, type, and the duration of injury. HybHPs are located in the periportal area and express progenitor markers, such as *Sox9*, and hepatocyte markers, such as *Hnf4α*. This population of specialized hepatocytes are able to reconstitute the liver mass after various chronic injuries[18]. *Sox9+/Hnf4α+* bi-phenotypic hepatocytes able to differentiate in either hepatocytes or cholangiocytes have been also identified in DDC-injured livers[43]. These cells represent the intermediate status of lineage conversion during liver regeneration. We found that conditional deletion of TFEB in the liver results in impaired LPCs activation. Moreover, we demonstrated that TFEB depleted hepatocytes fail to express *Sox9* after liver damage, thus suggesting that TFEB-dependent *Sox9* induction plays an important role in liver homeostasis in response to injury. This observation is in line with the known role of TFEB in the cellular adaptation to environmental cues such as various types of stress conditions[30,49].

It is known that overexpression of endogenous MiT-TFE transcription factors, as a result of chromosomal abnormalities such as translocations or gene amplifications, can drive tumorigenesis in renal cell carcinoma and melanoma[31,32,34,57] and to support cancer growth in pancreatic cancer[32–34]. However, a specific role for TFEB in cell differentiation, proliferation and neoplastic transformation in the liver has never been

demonstrated. Here, we show that *Albumin*-CRE-driven TFEB overexpression in the liver of transgenic mice results in a severe phenotype characterized by a progressive increase in liver mass, hyperplasia of bile ducts, altered biliary tree structure, multifocal biliary cystic hyperplasia, and fibrosis. Cholestasis, bile duct proliferation, and cystic alterations rapidly progress ultimately leading to biliary tumor with features resembling CCA. Previous studies showed that SOX9 is highly upregulated in many premalignant lesions and in tumor tissues and plays a role in tumor development[58–60]. Our results suggest that TFEB-mediated upregulation of *Sox9* expression plays a role in liver cancer development by inducing progenitor cell proliferation.

In conclusion, our study identifies an important role for TFEB in liver development and cell fate under physiological and pathological conditions. Further studies of this mechanism may lead to the identification of therapeutic targets for the modulation of liver regeneration after injury.

## Methods

**Mouse experiments.** Experiments performed in this study were conducted in accordance with the Baylor College of Medicine Institutional Animal Care and Use Committee (IACUC) with an approved protocol for the care and maintenance of laboratory animals. Mice analyzed for the study were males to reduce variability and were maintained on a C57BL/6 strain background. Housing conditions were as follows: temperature set point is 72 °F (22.2 °C), light cycle of 14 h and dark cycle of 10 h. Standard food and water were given ad libitum.

**TFEB overexpression in hepatocytes**

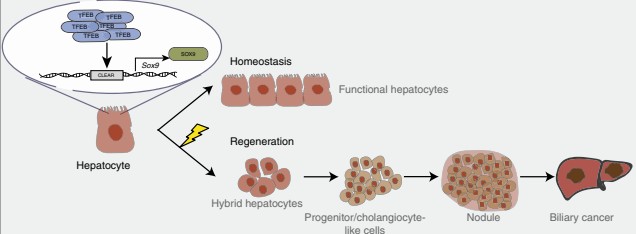

**TFEB overexpression in cholangiocytes**

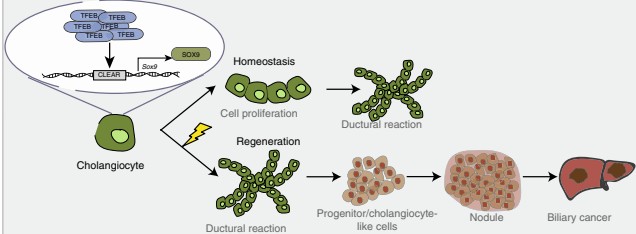

**TFEB ko in hepatocytes and cholangiocytes**

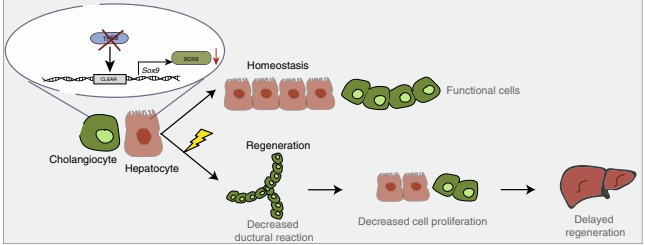

**Fig. 9 Schematic representation of the effects of TFEB on liver cell differentiation in vivo.** TFEB overexpression in hepatocytes does not affect hepatocyte proliferation and identity in homeostasis conditions, while it induces differentiation of hepatocytes into ductular reaction (DR)-associated cholangiocytes after injury. Conversely, TFEB overexpression in cholangiocytes induces their proliferation and DR in homeostasis and upon liver injury. On the other hand, TFEB depletion in the liver impairs normal cell regeneration after injury due to reduced induction of *Sox9* expression and decreased cell proliferation.

*Tcfeb*[LacZ/LacZ], *Tcfeb*[flox/flox], and T*cfeb*-3xFlag[fs/fs] transgenic mouse line generation has been previously described[25,26]. Wild type, *Sox9*[flox/flox], *R26R*[LSL]*tdTomato*, *Albumin*-Cre, and *Krt19*[CreERT] mouse lines were obtained from the Jackson laboratory (Bar, Harbor, ME).

The *Krt19*[CreERT] was induced by three individual intraperitoneal injections of tamoxifen at a dose of 4 mg. *Krt19*[CreERT] mice received 2 weeks of normal diet after the last tamoxifen injection before starting an injury diet regime.

To induce liver injury, mice were given 0.1% DDC food (Custom Animal Diet, Bangor, PA) as indicated in the text. After injury, mice were given normal chow and drinking water.

**Cell culture.** HBs were prepared from control and T*cfeb*-3xFlag[fs/fs] mice at embryonic day 13.5 and immortalized by plating at clonal density[61]. HBs were maintained in HB media (RPMI) (Thermo Fisher Scientific, Wilmington, DE) containing 10% FBS, 1% penicillin–streptomycin, 50 ng/ml epidermal growth factor, 30 ng/ml insulin-like growth factor II (PeproTech, Rocky Hill, NJ), 10 μg/ml insulin (Roche Diagnostics, Indianapolis, IN) on plates coated with rat tail collagen (BD Bioscience, San Jose, CA) in a humidified atmosphere with 5% $CO_2$ at 37 °C. Cells were infected with HDAd-BOS-CRE or HDAd-CMV-LacZ vectors (previously described[62,63]) and then cultured for the specific assay. For hepatocyte differentiation, HBs were cultured on uncoated tissue culture dishes in HB medium for up to 5 days. For bile duct differentiation, 6 cm tissue culture dishes were coated with 0.5 ml Basement Membrane Matrix (BD Bioscience, San Jose, CA) and allowed to set for 1 h. HBs in HB cell media supplemented with 100 ng/ml recombinant mouse hepatocyte growth factor (R&D System, Minneapolis, MN) were then added to the plate. Tubule formation was monitored at 24 h, 3 days, and 10 days.

For primary mouse hepatocytes isolation, we used a two-step perfusion technique previously described[64]. Briefly, WT or transgenic mouse liver was perfused with collagenase (C5138; Sigma-Aldrich, Saint Louis, MO) and digested to extract parenchymal cells out from the liver. The hepatocyte suspension was further purified using a 40% of Percoll gradient (P4937; Sigma-Aldrich, Saint Louis, MO), washed with Hepatocyte Wash Medium (17004-024; Thermo Fisher Scientific), and seeded at an appropriate cell density. Cells were harvested for analysis after 24 h.

**Generation of *Tcfeb*[KO] HBs through CRISPR-Cas9.** To generate TFEB[KO] HBs, we designed and synthetized two sgRNAs targeting adjacent regions of *Tcfeb* exon 3 as previously described[65,66] with minimal modification. CRISPRscan[67] was used to design forward primers containing the T7 promoter sequence, the proto-spacer sequence, and the sgRNA scaffold overlap sequence. Full-length sgRNA scaffold was obtained through an overlap PCR with the universal scaffold reverse primer.

Tfeb sgRNA 1 Forward:
taatacgactcactataGGGTATCTGTCTGAGACCTAgttttagagctagaaatagc
Tfeb sgRNA 2 Forward:
taatacgactcactataGGCAGGCTTCGGGGAACCTTgttttagagctagaaatagc
Universal scaffold Reverse:
gttttagagctagaaatagcaagttaaaataaggctagtccgttatcaacttgaaaaagtggcaccgagtcggtgct

In vitro transcription (IVT) was performed using the HiScribe T7 High Yield RNA Synthesis kit (NEB) following the manufacturer's instructions. A pre-complex of purified sgRNAs and *sp*Cas9 protein (IDT) was generated and then sgRNA-Cas9 ribonucleoproteins (RNP) containing the two sgRNAs were simultaneously transfected into primary mouse HBs using the NEON Transfection System (Thermo Fisher). The electroporation conditions were: Buffer R, 1400 V, 10 ms, three pulses. Deletion efficiency was assessed through PCR using the following primers:

*Tcfeb* Forward: GTCCACTTCCAGTCGCCC;
*Tcfeb* Reverse: AGGCTAGAGGCCCATAAAGAA.

**Flow cytometry and cell cycle analysis.** HBs were cultured on collagen-coated plates, harvested with trypsin-EDTA (Thermo Fisher Scientific, Wilmington, DE), and washed in PBS. For cell cycle analysis, cells were fixed in 1 ml cold 70% ethanol for 30 min on ice and then centrifuged for 5 min at high speed. The pellet was washed twice in PBS and treated with RNAseA (Life Technologies, Carlsbad, CA). Propidium iodide (PI) solution was added to the cells and the mixture incubated for 10 min at room temperature. Cell debris and dead cells were excluded from FSC-A/SSC-A dot-plots (Supplementary Fig. 10a). DNA content and relative cell cycle phases were determined based on PI relative fluorescence levels on singlets (Supplementary Fig. 10b). FACS experiments were performed using an LSRII cytometer (BD Bioscience, San Jose, CA Bioscience, San Jose, CA). Data were analyzed using FACSDiva (BD Bioscience, San Jose, CA) and Flow-Jo (Flow-Jo, LLC, Ashland, OR) software. Cell cycle phases were quantified using the Flow-Jo Cell Cycle platform (univariate). In all experiments at least 10,000 events were acquired.

**RNA extraction and quantitative RT-PCR.** Total RNA was extracted from livers using TRIzol reagent (Life Technologies, Carlsbad, CA) followed by the RNeasy kit (Qiagen, Hilden, Germany). RNA from cellular lysate was extracted using RNeasy kit according to the manufacturer's instructions. cDNA was synthesized by reverse transcription using a first-strand complementary deoxyribonucleic acid kit with random primers (Applied Biosystems) according to the manufacturer's protocol. The generated cDNA was diluted and used as a template for RT-PCR reactions, which was performed using the CFX96 Real Time System (Bio-Rad, Hercules, CA). Melting curve analysis were performed to verify amplification specificity. The PCR reaction was performed using iTaq SYBR Green Supermix (Bio-Rad, Hercules, CA) with the following thermocycler conditions: pre-heating, 5 min at 95 °C; cycling, 40 cycles of 15 s at 95 °C, 15 s at 60 °C, and 25 s at 72 °C. Relative quantification of gene expression was performed according to the 2 $(-\Delta\Delta CT)$ method. The *ß2-microglobulin*, *Ribosomal protein S16* or *Cyclophilin* genes were used as endogenous controls (reference markers). Primers used for RT-PCR are listed in Supplementary Table 6.

**Western blotting.** Liver and cell samples were homogenized in ice-cold lysis buffer (50 mM Tris–HCl pH 7.4, 150 mM NaCl, 1% Triton X-100, 1 mM EDTA pH 8.0, 0.1% SDS) supplemented with complete protease and phosphatase inhibitors (Roche Diagnostics, Indianapolis, IN). Samples were kept for 20 min at 4 °C and centrifuged at 16,000*g* for 10 min. Protein concentration was measured by BCA assay (Thermo Fisher Scientific). Denaturated samples were resolved on a 4–20% SDS-PAGE polyacrylamide gel by electrophoresis, transferred to a PVDF membrane, and then exposed to primary antibody overnight at 4 °C. Antibodies and dilution used for immuno-blots are listed in Supplementary Table 7. Uncropped blots are present in Supplementary Figs. 11 and 12.

**Cellular fractionation.** Enriched nuclear and cytosolic cellular subfractions were isolated by differential centrifugation, as previously described[47]. Liver was homogenized using a Teflon pestle and suspended in cytosol isolation buffer (250 mM sucrose, 20 mM HEPES, 10 mM KCl, 1.5 mM MgCl₂, 1 mM EDTA, 1 mM EGTA) supplemented with protease and phosphatase inhibitors (Roche Diagnostics, Basel, Switzerland). The lysate was then centrifuged at 1000 *g* for 10 min at 4 °C to pellet

the nuclei while the supernatant (cytosol) was re-centrifuged twice at $16,000g$ for 20 min at 4 °C and the pellets were discarded. Nuclei were re-suspended in nuclear lysis buffer (1.5 mM $MgCl_2$, 0.2 mM EDTA, 20 mM HEPES, 0.5 M NaCl, 20% glycerol, 1% Triton X-100), incubated on ice for 30 min, and then sonicated 3 Å ~10 s followed by a final centrifugation step of 15 min at $16,000g$. The enriched nuclear and cytosolic fractions were analyzed by Western blot analysis.

**In situ hybridization**. Livers from P9 and 3-month-old WT and Tg mice were collected and frozen without prior fixation in OCT cryoprotection media. Tissue was cut into 25-μm sections, mounted on slides with ProlongGold with DAPI (Life Technologies, Carlsbad, CA), and used for RNA in situ hybridization. We utilized the previously described method[68] but modified the development of the signal by using a Cy3-labeled tyramide instead of the biotin-labeled tyramide. *Tcfeb*-probe-F1 5′-GCGGCAGAAGAAAGACAATC-3′ and *Tcfeb*-probe-R1 5′-AGGTGATG-GAACGGAGACTG-3′ were used to amplify 1300 bp from mouse liver cDNA and cloned into pGEM_T Easy vector (Promega, Madison, MI). This template was used to generate a DIG-labeled mRNA probe using IVT reagents from Roche (Roche Diagnostics, Indianapolis, IN).

**Liver staining**. Dissected livers were processed for paraffin embedding as previously described[47]. Paraffin blocks were cut into 6 μm sections. For Sirius red staining the sections were rehydrated and stained for 1 h in picro-sirius red solution (0.1% Sirius red in a saturated aqueous solution of picric acid) (Sigma-Aldrich, Saint Louis, MO). After two changes of acidified water (5 ml glacial acetic acid in 1 l of water), the sections were dehydrated in three washes of 100% ethanol, cleared in xylene, and mounted on a resinous medium. X-Gal and H&E stainings were performed following the IHC World protocol. For immunostaining, the sections were rehydrated to PBS pH 7.4 and permeabilized with 0.2% Triton in PBS. Heat Induced Epitope Retrieval using the citrate buffer method (pH 6.0) was performed to retrieve the antigen sites. The sections were then incubated for 1 h at room temperature with blocking solution (3% BSA, 5% donkey serum, 20 mM $MgCl_2$, 0.3% Tween 20 in PBS pH 7.4). For co-stainings, primary antibodies were incubated overnight at 4 °C. Secondary antibodies made in donkey were: AlexaFluor-488 anti-rabbit and AlexaFluor-555 anti-mouse (Life Technologies, Carlsbad, CA). Slides were mounted on ProlongGold with DAPI (Life Technologies, Carlsbad, CA). For IHC, the Vectastain ABC kit and DAB (Vector Laboratories, Burlingame, CA) were used following the manufacturer's instructions. Sections were counterstained using Mayer's hematoxylin (Electron Microscopy Sciences, Hatfield, PA). For BrdU staining, 2 mg of BrdU (BD Bioscience) diluted in PBS 1× was injected by intraperitoneal injection 4 h before tissue collection. Livers were dissected, fixed, with buffered 10% formalin overnight at 4 °C and embedded into OCT blocks and cut into 10-μm sections. BrdU signal was revealed by using a rat anti-BrdU antibody. Stained liver sections were examined under a Zeiss Axiocam MR microscope and analyzed using Zen black software. For staining quantification, Image J Software (NIH) was used to calculate the percent area positively stained in five random low power views.

**Gene expression analysis**. Total RNA from HBs 3 days after hepatocytic differentiation was quantified using the Qubit 2.0 fluorimetric Assay (Thermo Fisher Scientific). Libraries were prepared from 100 ng of total RNA using the QuantSeq 3′ mRNA-Seq Library Prep Kit FWD for Illumina (Lexogen GmbH). Quality of libraries was assessed by using screen tape High sensitivity DNA D1000 (Agilent Technologies). Libraries were sequenced on a NovaSeq 6000 sequencing system using an S1, 100 cycles flow cell (Illumina Inc.). Sequence reads were trimmed using bbduk software (bbmap suite 37.31) to remove adapter sequences, poly-A tails, and low-quality end bases (regions with average quality below 6). Alignment was performed with STAR 2.6.0a on mm10 reference assembly. The expression levels of genes were determined with htseq-count 0.9.1 by using mm10 Ensembl assembly (release 90). We have filtered out all genes having <1 cpm in less than two samples. Differential expression analysis was performed using edgeR[69].

Microarray analysis was performed on liver from P9 mice of each of the two genotypes ($n = 3$ per group). All samples were processed on Affymetrix Mouse 430A 2.0 arrays using GeneChip 3′-IVT Plus and Hybridization Wash and Stain kits by means of Affymetrix standard protocols. Raw intensity values of the six arrays were processed and normalized by Robust Multi-Array Average Method[70] using the Bioconductor R package Affy[71]. The data have been deposited in NCBIs Gene Expression Omnibus[72] (GEO) and are accessible through GEO Series accession number GSE35015.

Enrichment analyses of KEGG pathways on differentially expressed genes were performed using the Bioconductor R package clusterProfiler (FDR ≤ 0.05). Gene GSEA was run on HNF4α targets downloaded from the Molecular Signatures Database (MSigDB)[73].

**Promoter analysis**. The analysis of TFEB binding sites was performed using the CLEAR matrix[74] and the matchPWM algorithm implemented in the Biostrings package. Putative binding sites were filtered with threshold of significance set at 0.8.

**Chromatin immunoprecipitation**. ChIP was performed using livers of 2-month-old Alb-Cre;*Tcfeb*-3xFlag and control mice as previously described[30]. Primers used for RT-PCR are listed in Supplementary Table 8.

**Luciferase assay**. The *Sox9* promoter was amplified from mouse genomic DNA and cloned into the pGL4 plasmid (Promega, Madison, MI). A quick-change site directed mutagenesis kit (Agilent, Santa Clara, CA) was used for the mutagenesis of CLEAR sites in the *Sox9* promoter. Primers used for mutagenesis are listed in Supplementary Table 9. The *Sox9* promoter-reporter luciferase construct was transfected along with the pRL-TK (Promega, Madison, MI) in control HBs and HBs overexpressing *Tcfeb*. Twenty-four hours after transfection cells were collected and subjected to luminescence detection by using the Dual-Luciferase Reporter Assay system (Promega, Madison, MI). The luminescence was measured using a Turner Designs Luminometer (DLReady) (Promega, Madison, MI) and normalized against *Renilla* luciferase activity.

**Biliary tree casting**. Two-month-old mice were euthanized with isoflurane (Vedco Saint Joseph MO). Ink (Hyatt's) was injected into the common bile duct using a 30-gauge needle. The entire liver was removed, formalin-fixed, and clarified in a 1:2 solution of benzyl alcohol and benzyl benzoate.

**Blood chemistry analysis**. Blood samples were collected by retroorbital bleeding. Serum was frozen at −80 °C or used immediately after collection. Total Bile Acids (TBA) Enzymatic Cycling Assay Kit was obtained from BQKITS (San Diego, CA).

**Vector production and injections**. pAAV-TBG-GFP-pA (GFP) was obtained from Thomas Vallim (University of California, Los Angeles). pAAV-TBG-PI-Cre-rBG (Cre Recombinase) and plasmids required for AAV packaging, adenoviral helper plasmid pAdDeltaF6 (PL-F-PVADF6), and AAV8 packaging plasmid pAAV2/8 (PL-T-PV0007) were obtained from the University of Pennsylvania Vector Core. AAV were generated as previously described with some modifications[75,76]. Each AAV transgene construct was co-transfected with the packaging constructs into 293 T cells (ATCC, CRL-3216) using polyethylenimine (PEI). Cell pellets were harvested and purified using a single cesium chloride density gradient centrifugation. Fractions containing AAV vector genomes were pooled and then dialyzed against PBS using a 100kD Spectra-Por® Float-A-Lyzer® G2 dialysis device (Spectrum Labs, G235059) to remove the cesium chloride. Purified AAV were concentrated using a Sartorius™ Vivaspin™ Turbo 4 Ultra-filtration Unit (VS04T42) and stored at −80 °C until use. AAV titers were calculated after DNase digestion using the qPCR standard method. Primers used for titer are included in Supplementary Table 10. Viruses were administered by retroorbital injection at the dose of $2 \times 10^{11}$ genome copies/mouse.

**Statistical analyses**. Data are expressed as averages ± standard error. Statistical significance was computed using GraphPad Prism v7 using Student's two-tailed *t*-test or two-way ANOVA as indicated in the figure legends. A *p* value <0.05 is considered statistically significant.

**Reporting summary**. Further information on research design is available in the Nature Research Reporting Summary linked to this article.

## Data availability

The data that support the findings of this study are available from the corresponding authors upon reasonable request.

Sequencing data have been deposited in the NCBIs Gene Expression Omnibus (GEO) database under the accession code GSE35015.

The source data underlying Figs. 2b, c, f, 3c, 4a–d, 5b–d, f, 6b, 7b, 8c, e, Supplementary Figs. 1c, d, 2e, 3c, 4a, 8a–h, 9a, d are provided as a Source Data file.

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

## Acknowledgements

We thank Huda Zoghbi, Nicola Brunetti-Pierri, Roberto Zoncu, Rushika Perera, and Graciana Diez Roux for critical reading of the manuscript. We are grateful to Diego Carrella and Rossella De Cegli (TIGEM) for promoter analysis and ChIP primer design. We thank Dr. Thomas Vallim (University of California, Los Angeles) for the pAAV-TBG-GFP plasmid. We also thank the NGS Core at TIGEM for help with gene expression analysis. This work was supported by a grant from the US National Institutes of Health (R01-NS078072), AIRC (Italian Association for Cancer Research) (IG 2015-17639), Foundation Louis-Jeantet and Telethon Foundation to A.B. The project was supported in part by the RNA In Situ Hybridization Core facility at Baylor College of Medicine which is supported by a Shared Instrumentation grant from the NIH (1S10OD016167) and the NIH IDDRC Grant (1U54HD083092-02) from the Eunice Kennedy Shriver National Institute of Child Health & Human Development. This project was also supported in part by the NIDDK Digestive Disease Center Core with funding from the NIH (NIH P30DK58338) and in part by the Pathology and histology Core at Baylor College of Medicine with funding from the NIH (NCI-CA125123).

## Author contributions

N.P. performed the experiments, interpreted the results, and generated the Fig.9. T.H., N.J.H., A.C., and L.D. provided technical support to N.P. T.J.K. performed ChIP experiments. L.B. helped with the generation of CRISPR/Cas9 clones and with FACS analysis. K.K. isolated primary hepatocytes. M.D.G., A.H., and W.R.L. provided AAV8 viruses. A.C. and M.M. analyzed the gene expression data. N.A., D.D.M., C.S., M.J.F., and S.J.F. contributed to interpretation of the results. A.B. and N.P designed the overall study, supervised the work, and wrote the manuscript.

## Competing interests

A.B. is a co-founder of CASMA Therapeutics, Boston, MA, USA. The remaining authors declare no competing interests.
