## [Peer Review File · Nature Communications]

Reviewers' Comments:

Reviewer #2:

Remarks to the Author:

The authors have sufficiently addressed all the concerns that I raised in previous review round. I think the manuscript is considered to be acceptable for publication in Nature Communications now.

Reviewer #3:

Remarks to the Author:

I thank the authors for addressing all my comments. The lineage tracing experiments that specifically map the fate of either TCFEB overexpressing hepatocytes or cholangiocytes in homeostasis and following injury are key. The conclusion is clear and demonstrates that OE of TCFEB in cholangiocytes induce their proliferation and DR development in homeostasis and following DDC injury. OE in hepatocytes doesn't affect hepatocyte proliferation in homeostasis conditions, yet induces differentiation of hepatocytes into DR-associated cholangiocytes after DDC injury. However, surprisingly, the authors have not tested whether OE of TCFEB in cholangiocytes will still prevent the conversion of cholangiocytes into hepatocytes when the proliferation of hepatocytes is compromised (following OE of p21 in hepatocytes as published by Dr. Forbes who is a co-author in this study). This data is missing.

I would recommend redoing the cartoon in Supp Fig 10, as it is confusing as it is. Since the authors have elegantly dissected the effects of OE and KO in cholangiocytes and hepatocytes separately, these data from both cell types should be clearly represented here.

Reviewers' comments:

Referee #2

The authors have sufficiently addressed all the concerns that I raised in previous review round. I think the manuscript is considered to be acceptable for publication in Nature Communications now.

We are grateful to reviewer 2.

Referee #3

I thank the authors for addressing all my comments. The lineage tracing experiments that specifically map the fate of either TCFEB overexpressing hepatocytes or cholangiocytes in homeostasis and following injury are key. The conclusion is clear and demonstrates that OE of TCFEB in cholangiocytes induce their proliferation and DR development in homeostasis and following DDC injury. OE in hepatocytes doesn't affect hepatocyte proliferation in homeostasis conditions yet induces differentiation of hepatocytes into DR-associated cholangiocytes after DDC injury. However, surprisingly, the authors have not tested whether OE of TCFEB in cholangiocytes will still prevent the conversion of cholangiocytes into hepatocytes when the proliferation of hepatocytes is compromised (following OE of p21 in hepatocytes as published by Dr. Forbes who is a co-author in this study). This data is missing.

We have performed lineage tracing upon overexpression of TFEB in cholangiocytes and in hepatocytes in homeostasis and after injury. As the reviewer says, "these experiments are key and the conclusion is clear". We believe that the results of these experiments provide solid evidence on the role of TFEB in modulating liver cell fate. Based on these considerations, we believe that the experiments performed by blocking the proliferation of hepatocytes (i.e. following OE of p21 in hepatocytes) are not necessary to support the main points of the paper.

I would recommend redoing the cartoon in Supp Fig 10, as it is confusing as it is. Since the authors have elegantly dissected the effects of OE and KO in cholangiocytes and hepatocytes separately, these data from both cell types should be clearly represented here.

We thank the reviewer for suggesting to modify Supplementary Fig 10 (now Fig.9). We follow the reviewer's suggestion by representing both OE and KO data obtained in cholangiocytes and hepatocytes.